# Loss of Blood-Brain Barrier Integrity in an In Vitro Model Subjected to Intermittent Hypoxia: Is Reversion Possible with a HIF-1α Pathway Inhibitor?

**DOI:** 10.3390/ijms24055062

**Published:** 2023-03-06

**Authors:** Anne Cloé Voirin, Morgane Chatard, Anne Briançon-Marjollet, Jean Louis Pepin, Nathalie Perek, Frederic Roche

**Affiliations:** 1INSERM, SAINBIOSE U1059, Univ. Jean Monnet Saint-Étienne, Mines Saint Etienne, F-42023 Saint-Étienne, France; 2Univ. Grenoble Alpes, Inserm, CHU Grenoble Alpes, HP2 Laboratory U1300, 38000 Grenoble, France

**Keywords:** blood-brain barrier, tight junction, ABC transporters, intermittent hypoxia, HIF-1

## Abstract

Several sleep-related breathing disorders provoke repeated hypoxia stresses, which potentially lead to neurological diseases, such as cognitive impairment. Nevertheless, consequences of repeated intermittent hypoxia on the blood-brain barrier (BBB) are less recognized. This study compared two methods of intermittent hypoxia induction on the cerebral endothelium of the BBB: one using hydralazine and the other using a hypoxia chamber. These cycles were performed on an endothelial cell and astrocyte coculture model. Na-Fl permeability, tight junction protein, and ABC transporters (P-gp and MRP-1) content were evaluated with or without HIF-1 inhibitors YC-1. Our results demonstrated that hydralazine as well as intermittent physical hypoxia progressively altered BBB integrity, as shown by an increase in Na-Fl permeability. This alteration was accompanied by a decrease in concentration of tight junction proteins ZO-1 and claudin-5. In turn, microvascular endothelial cells up-regulated the expression of P-gp and MRP-1. An alteration was also found under hydralazine after the third cycle. On the other hand, the third intermittent hypoxia exposure showed a preservation of BBB characteristics. Furthermore, inhibition of HIF-1α with YC-1 prevented BBB dysfunction after hydralazine treatment. In the case of physical intermittent hypoxia, we observed an incomplete reversion suggesting that other biological mechanisms may be involved in BBB dysfunction. In conclusion, intermittent hypoxia led to an alteration of the BBB model with an adaptation observed after the third cycle.

## 1. Introduction

Oxygen is a key component of brain function. The brain is the first consumer of oxygen, using 20% of the arterial supply [1]. One of the protective structures of the brain known to be highly sensitive to hypoxic conditions is the blood-brain barrier (BBB) [2].

The BBB is composed of endothelial cells which constitute the main structure and function of this barrier. The first characteristic of the BBB is the low permeability associated with the tight junction proteins expressed by the endothelial cells, e.g., zonula occludens (ZO) and/or claudin (Cld) proteins. The second characteristic is the metabolic function mainly represented by the efflux pumps, including ABC transporters such as P-glycoprotein (P-gp) and multidrug resistance protein-1 (MRP-1), which confer the first line of defense of the brain [3,4].

In this context, many in vitro models of the BBB have been developed to understand the function of this barrier under physiological and/or pathological conditions such as hypoxia. A number of studies have evaluated the effect of acute hypoxic stress on BBB models which mimic stroke or cerebral ischemic stress [1,5,6,7]. All of these studies have shown that acute hypoxia leads to an alteration of the BBB, associated with a loss of expression of tight junction proteins and linked to an increase in permeability [8]. Although many studies have focused upon the consequences of acute hypoxic stress, there are few data on repeated hypoxic stress. Repeated hypoxic stress is found in many pathologies, including sleep related breathing disorders, such as obstructive sleep apnea syndrome (OSAS) or obesity hypoventilation syndrome. In the context of OSAS, intermittent hypoxia (IH) becomes chronic, initiating and amplifying structural alterations of the brain. Studies have shown that this repeated hypoxia does not appear to be harmless for the brain, and cognitive deficits observed in these patients are the consequence of a possible modification of cellular homeostasis [9,10,11]. In addition, hypertrophic modifications of the grey matter and white matter have been observed in different cerebral regions by focal volumetric imaging [12,13], leading to possible neuroinflammation and acute cerebral hyperpermeability [14]. A study has shown that a variation of brain aquaporin after IH in a mouse model suggested a link with modification of the BBB [15]. Presently, the best treatment for OSAS is continuous positive airway pressure (CPAP), a device associated with a mask delivering an airway pressure support oxygen continuously through the night [16]. This treatment has shown its capacity to improve cognitive functions, such as attention and memory, in the context of OSA [17,18]. Unfortunately, this treatment is poorly tolerated and often abandoned by OSAS patients. Moreover, some OSAS patients remain at risk of cerebrovascular dysfunction in spite of CPAP treatment. Hence it is of great interest to develop new therapies that are better tolerated and accepted by OSAS patients. In the context of OSAS, the challenge is to have a better understanding of the mechanisms involved in cognitive impairment in order to develop new therapies. One of the recently developed hypotheses explaining the neurological consequences of OSAS is the alteration of the permeability and function of the BBB via the Hypoxia Inducible Factor-1 (HIF-1) pathway stabilization [19]. Studies have shown that cellular hypoxia promotes a signal, which upon detection of an insufficient supply of oxygen will lead to the translocation and dimerization of HIF-1 subunits into a functional complex, thus increasing the nuclear activation of HIF-1 target genes [20,21,22]. Prabhakar et al. (2012) showed that IH associated with sleep disordered breathing leads to HIF-1 imbalance, linked to cardiorespiratory pathology [23] and hypertension found in OSAS [24]. In vivo studies have shown the link between IH, HIF-1 and reactive oxygen species (ROS) in cardiorespiratory changes [25]. This HIF-1 pathway thus appears to be an important component in cardiorespiratory dysfunction. It could therefore also be linked to cerebral dysfunction as suggested by Lim and Pack [26]. 

In a previous study, our laboratory developed a chemical method for the induction of repeated hypoxia, allowing activation of the HIF-1 pathway, and thus its evaluation in in vitro models of the BBB. This method of induction of hypoxia was rendered possible by the use of a chemical agent, hydralazine. This agent is a prolylhydroxylase domain inhibitor and has been validated in previous work in inducing a signaling pathway similar to those activated by hypoxia by inducing overexpression of HIF-1α [27,28]. Hydralazine also shows a capacity to induce a transient and physiological HIF-1α overexpression by inhibiting PHD activity. In the literature, hydralazine was used to mimic a hypoxic state in cancer models and endothelial cells [29,30]. However, hydralazine does not induce a cellular hypoxic environment and it appears necessary to compare such a chemical stress with physical hypoxia.

One hypothesis is that hypoxia repeated stress has consequences on BBB permeability mediated HIF1 α. In this work, we evaluated the impact of repeated hypoxic stress mimicking pathological conditions and chronical stresses on an in vitro BBB model, inducing HIF-1α content by using either chemical hydralazine or an IH chamber that creates a cellular hypoxic environment alternating with a normoxic environment. Firstly, we evaluated the effect of these stresses on the paracellular pathway of the barrier model via the amount of tight junctions and permeability measurement, which are essential properties. Moreover, we evaluated the metabolic properties of the BBB by studying the abundance of ABC transporters necessary for the efflux of exogenous substances from the endothelial cells. Secondly, we evaluated whether inhibition of the HIF-1 α may prevent opening of the BBB, by reversal studies using 3-(5′-hydroxyethyl-2′-furyl)-1-benzylindazole (YC-1) a molecule previously shown to inhibit the expression of HIF-1 alpha in several other studies and in the context of acute hypoxia [31].

## 2. Results

### 2.1. Hydralazine and IH Induce HIF-1α Expression

bEnd.3 cells significantly overexpressed HIF-1α after 2 h exposure to hydralazine or after 2 h of IH (*p* < 0.01) (Figure 1). HIF-1α content significantly decreased in bEnd.3 cells 6 h after hypoxic stress (*p* < 0.01). At this time, the cells recovered to a baseline level of HIF-1α level. Finally, pretreating with YC-1 reversed hydralazine-induced HIF-1α over. YC-1 thus efficiently inhibited HIF-1 α activation. 

### 2.2. Hydralazine-Induced HIF-1α Pathway Is Involved in BBB Impairment

#### 2.2.1. Inhibition of HIF-1 Abolishes Hydralazine-Induced BBB Hyperpermeability

The apparent permeability (Papp) measurement (Figure 2A) showed a gradual increase in permeability after repeated stresses with hydralazine. After the first stress, Papp increased from 1.58 × 10^−6^ ± 0.08 cm·s^−1^ to 3.64 × 10^−6^ ± 0.09 cm·s^−1^. Permeability to Na-Fl significantly increased significantly up to 4.04 × 10^−6^ ± 0.08 cm·s^−1^ after the second stress, and to 4.97 × 10^−6^ ± 0.17 cm·s^−1^ after the third stress. Inhibition of HIF-1 with YC-1 maintained this permeability similar to that found in normoxic condition. The Papp was 1.78 × 10^−6^ ± 0.04 cm·s^−1^ after the first stress, 1.82 × 10^−6^ ± 0.06 cm·s^−1^ after the second stress, and 1.98 × 10^−6^ ± 0.17 cm·s^−1^ after the third stress.

Similarly, repeated hydralazine stress altered transendothelial electrical resistance (TEER) in our in vitro BBB model. TEER values significantly decreased from 87.4 ± 4.63 to 59.4 ± 1.5 Ω·cm^2^ after the first stress, to 58.8 ± 1.4 Ω·cm^2^ after the second stress, and finally to 60.0 ± 1.2 Ω·cm^2^ after the third stress, which represented a 30% decrease (*p* < 0.05). Reversal studies with YC-1 showed that TEER slightly diminished compared to the hydralazine alone condition from 87.4 ± 4.6 Ω·cm^2^ to 71.9 ± 1.3 Ω·cm^2^ after the first stress, to 76.6 ± 1.0. Ω·cm^2^ after the second stress, and to 76.3 ± 1.2 Ω·cm^2^ after the third stress (*p* < 0.05 vs. conditions without YC-1).

#### 2.2.2. YC-1 Reverses Hydralazine-Induced Alterations in Expression of Tight Junction Proteins

We measured two tight junction protein levels: ZO-1 and claudin-5 in our in vitro BBB model after repeated hydralazine-induced stress in the presence or absence of YC-1 (Figure 3). ZO-1 content progressively decreased after each hydralazine-induced stress. After the first stress, ZO-1 level significantly decreased by 26%, then by 60% after the second stress, and by 72% after the last stress (*p* < 0.05). YC-1 reversed the effects of hydralazine on ZO-1 abondance expression after the second and third cycles. The level of claudin-5 dramatically decreased after repeated hydralazine-induced stress. We noticed a significant decrease by 37% after the first stress, by 71% after the second stress, and by 75% after the last exposure to hydralazine (*p* < 0.05). Claudin-5 concentration was maintained after each repeated hydralazine stress with YC-1 treatment, with only weak decreases by 4% to 10% after the different hydralazine exposures.

#### 2.2.3. YC-1 Reverses Hydralazine-Induced Overexpression of ABC Efflux Transporters Proteins

We investigated the content of P-gp and MRP-1, two ABC transporters proteins in our in vitro BBB model after the three repeated hydralazine-induced stresses (Figure 4). MRP-1 concentration significantly increased after each hydralazine-induced stress; from 5.67 ± 1.22 to 7.51 ± 1.11 µg/mL after the first exposure, to 9.31 ± 1.34 µg/mL after the second exposure, and up to 11.07 ± 0.61 µg/mL after the third exposure. MRP-1 protein level was maintained at its basal level with YC-1 pretreatment. Similarly, P-gp concentration gradually increased after each hydralazine exposure. Moreover, P-gp was down-regulated by almost 25% after the second and third exposures to hydralazine with YC-1.

### 2.3. IH-Induced HIF-1α Pathway Is Involved in BBB Impairments

#### 2.3.1. YC-1 Reverses IH-Induced Hyperpermeability

After IH, the Papp measurement evaluated by Na-Fl BBB passage showed an increase in permeability (Figure 5A). After the first stress, Papp to Na-Fl increased from 1.58 ± 0.08 cm·s^−1^ to 2.78 ± 0.17 cm·s^−1^, and after the second IH period it increased to 2.60 ± 0.11 cm·s^−1^. Interestingly, after the third IH exposure the Papp returned to control normoxic level, i.e., 1.71 ± 0.72 cm·s^−1^. When cells were preincubated with YC-1, the Papp was 2.24 ± 0.15 cm·s^−1^ after the first exposure and 2.11 ± 0.22 cm·s^−1^ after the second exposure, showing a reversion of IH effects on Papp by YC-1. Similarly, a significant decrease in TEER was observed after the first two stresses (Figure 5B). TEER values significantly decreased from 97.01 ± 3.63 to 83.49 ± 0.47 Ω·cm^2^ after the first IH period, and from 97.01 ± 3.63 to 86.79 ± 3.27 Ω·cm^2^ after the second IH period. Finally, TEER was maintained to control TEER values after the third IH period. YC-1 reestablished TEER values at the control level after the first IH period, and there was no significant difference to the control after the second and third IH periods.

#### 2.3.2. YC-1 Reversed IH Impact on the Level of Tight Junction Proteins

Determination of tight junction protein content (ZO-1 and claudin-5) in our in vitro BBB model was established after IH with and without YC-1 treatment (Figure 6). ZO-1 concentration significantly decreased by 35% after the first exposure, by 43% after the second exposure, and by 42% after the third exposure (*p* < 0.05). YC-1 significantly increased ZO-1 expression after each IH period compared to the cells without YC-1.

Similarly, the content of claudin-5 progressively decreased by 32% to 63% compared to normoxic condition after the different IH periods. Again, claudin-5 expression was reversed by YC-1 treatment (Figure 6).

#### 2.3.3. YC-1 Reversed IH Impact on the Level of ABC Efflux Transporter Proteins

Content of P-gp and MRP-1 showed a significant increase after the second and third exposures to IH. P-gp expression increased by 31% after the second exposure and by 58% after the third exposure; while MRP-1 level increased after the same time exposures by 66% and 111%, respectively. Reversal studies with YC-1 showed that P-gp and MRP-1 expression returned to normoxia levels after the last exposure (Figure 7).

## 3. Discussion

Brain functioning is highly influenced by local partial oxygen pressure. However, many pathologies are associated with local cyclical changes in oxygen partial pressure, such as sleep apnea syndrome. This pathology is characterized by chronic IH. The cerebrovascular and cognitive consequences associated with this syndrome have been well demonstrated [11,32,33], however, the underlying mechanisms and alterations are less known. Hypotheses are emerging and centered on HIF-1 expression that would contribute contributes to the increase in brain permeability [1,19].

The aim of our study was to improve knowledge on the post-hypoxic cellular risks associated with OSA, as well as to better understand the cellular and molecular mechanisms that would be involved during repeated hypoxia, such as the HIF-1 pathway. In this work we also proposed to evaluate therapeutic strategies centered on HIF-1α by using YC-1 inhibitor. Tsui et al. showed that use of YC-1 reduced the HIF-1α level and affected partially hypoxia- induced gene expression [34].

### 3.1. Prevention of BBB Impairment by Inhibition of HIF-1α with Hydralazine

Our previous studies with hydralazine in acute hypoxic stress have shown an increase in Na-Fl permeability associated with a decrease in the level of tight junction proteins such as ZO-1. In this present study, we confirmed these results and we further showed that this alteration of the endothelial characteristics of the BBB model is progressive along the repetition of the three stress cycles. In particular, we observed a gradual increase in apparent Na-Fl permeability, which was related to a progressive decrease in the content of tight junction proteins, such as ZO-1 and claudin-5. This opening implies the possible passage of potentially toxic compounds from blood to brain and vice versa. In parallel, endothelial cell seemed to set up a defense and detoxification system by increasing the amount of P-gp and MRP-1 proteins. These progressive modifications can be interpreted as a protective mechanism put in place by the BBB to limit the consequences associated with the loss of barrier tightness and are observed similarly to those obtained in acute hypoxia stress induced by hydralazine in vitro [28]. Hydralazine is known to be an activator of the HIF-1 in vivo pathway by acting as a PHD inhibitor and making it possible to mimic a hypoxia-induced stimulation of downstream genes associated with the HIF-1 pathway [29]. Consistent with our results, studies have shown that HIF-1α activation could be correlated with a decrease in ZO-1 level in b.End.3 endothelial cells [35] or in the increase of P-gp expression in colon carcinoma cells [36].

In complement, we decided to examine the preventive effect of an inhibitor of this HIF-1α pathway, YC-1. We chose YC-1 for its absence of cytotoxicity on b.End.3 and C6 cells and because it has been classically used to inhibit the HIF-1α pathway [35,37]. In adult rat brain endothelial cells subjected to ischemia-reperfusion, Yeh et al. showed an increase in permeability associated with ZO-1 disorganization when they associated ischemia and reperfusion conditions. Reversal studies with the use of YC-1 prevented the effects [37]. These results are in agreement with ours. The HIF-1α pathway was inhibited, we preserved our model integrity, and permeability and content of tight junction proteins were reversed close to that evaluated in normoxic condition. This result confirms that hydralazine involves, at least in part, the activation of the HIF-1 pathway and that YC-1 could be a therapeutical option. Nevertheless, intermittent hypoxic stress induced via hydralazine represents long hypoxic stresses (2 h hypoxia followed by 6 h normoxia), thus distinct from the pathophysiology found in OSAS. Hence, we wanted to compare the effect of this stress to an intermittent hypoxic stress induced by a specific chamber that allows shorter hypoxic cycles and is more in line with the pathophysiology of repeated OSA [38]. This novel device is able to expose multiple cell culture dishes to IH cycles relevant to OSA physiopathology. 

### 3.2. Prevention of IH-Induced BBB Impairment by Inhibition of the HIF-1 Pathway 

We observed an alteration in the integrity of endothelial cell function during the first and second cycles. This alteration was represented by an increase in apparent Na-Fl permeability and a decrease in TEER, correlating with a decreased content of ZO-1 and claudin-5, and a significant increase in amounts of ABC transporters P-gp and MRP-1. Similar results were found in hypoxia-reoxygenation models in the literature. Won et al. showed an increase in permeability, TEER and a modification of tightness structure with, in particular, a decrease in claudin-5 on b.End.3 cells [39]. Others have observed in vivo modifications in ischemia-reperfusion models with an increase in blue Evans permeability and a decrease in claudin-5 protein in the brain [40]. Finally, Mark et al. showed that an alteration in ZO-1 correlated with changes in permeability in a bovine brain microvascular endothelial cell model after hypoxia-reoxygenation cycles [41]. In addition, we showed an increase in HIF-1α expression which would produce similar changes [35,36], suggesting that IH effects may be mediated by HIF-1α activation. Interestingly, after the third cycle of IH we did not observe significant variations in permeability and TEER. Nevertheless, we observed an increased or unchanged content of ZO-1 and claudin-5 compared to the second cycle, and an increased concentration of P-gp and MRP-1 during the third cycle. This is consistent with the work of Dopp who reported an in vivo increase in P-gp mRNA after exposing rats to IH [42]. P-gp overexpression was also observed in rat brain microvascular endothelial cells subjected to IH in vitro [43]. Other studies report an increase in P-gp amount in an in vivo model after ischemia- reperfusion or in an in vitro model with immortalized endothelial cells after IH [44]. Ding et al. showed a positive correlation between HIF-1α and P-gp abondance in human colon carcinoma cells [36]. So, P-gp appears to be a key element in the regulation of BBB dysfunction during IH conditions, or upon activation of HIF-1α pathway. Few studies have investigated MRP-1 protein. We showed an original increase in its content. These original results suggest protection and adaptation processes developed by endothelial cells during this chronic IH exposure. In this context, Ibbotson et al. showed an increase of MRP-1 after hypoxia then reoxygenation and hypothesized that expression is regulated by transcription factor nuclear factor erythroid-2-related factor 2 (Nrf2) [45]. He suggested that a MRP isoform is useful for BBB protection in diseases involving hypoxia-reoxygenation. Furthermore, we performed reversal tests with a pretreatment of YC-1. We observed partial reversion of permeability, as well as other characteristics of the BBB, i.e., tight junction proteins and ABC transporter proteins. Thus, this incomplete reversion suggests that other mechanisms in addition to HIF-1α activation may be involved in repeated IH. In studies conducted in our laboratory with another model of IH (i.e., hypoxia (35 min, 1% O_2_)/reoxygenation (25 min, 18% O_2_)), we showed an increase of ROS production, as well as Nrf2 [46]. ROS are known to be involved in the opening of the BBB (28, 33) and to activate HIF-1α [21]. So it may be one step in HIF-1 activation, while Nrf2 is considered as a cytoprotective agent [47] and which may be one of the adaptive responses observed during the third cycle. Inflammatory pathways could also be involved in the deleterious effects on the barrier. We have shown in a previous study the deleterious effect of inflammation on our in vitro model of the BBB mediated by NFκB [48]. Consistently, we found many biomarkers such as inflammatory molecules or ROS in sera of patients suffering from OSAS [49,50] and their involvement on BBB opening [50]. 

Hypoxia-inducible factor 1 (HIF-1), and more specifically HIF-1α, is stabilized in response to hypoxia stress, and stimulates the transcription of genes involved in various biological processes. There are several pathways and it is difficult to determine which one is more implicated. HIF-1α exerts its effects through proteins coded by its downstream genes for example Protease Activated Receptor or *PAR-1*, *Vascular endothelial growth factor* (*VEGF*), and *glucose transporter* (*GLUT*). In our previous studies, we have seen that intermittent hypoxia provokes BBB disruption mediated PAR1 expression and reversal studies with PAR1 antagonist diminish BBB permeability linked to HIF expression [51]. We also demonstrated that IH induces ROS. Indeed, the production of ROS is known to alter BBB functioning mediated NRF2 transcription [46,51]. These downstream genes may express differently and exert different functions in different cell types. VEGF has been reported to have different effects on cell and tissue injuries. On the one hand, it might directly counteract the detrimental neurological effects associated with stroke. VEGF also promotes blood-brain barrier (BBB) permeability by altering tight junctions under ischemic and inflammatory conditions. Suppressing VEGF by HIF-1 inhibitors improves BBB permeability as observed by Yeh et al. [37]. Understanding cell-type dependent effects of HIF-1 will undoubtedly shed new lights on its role on BBB opening and provide potential approaches to promote its beneficial effect and reduce its detrimental function. The limitation of our study is to elucidate by which mechanism but we could not exclude that many pathways were involved as we have demonstrated in our previous papers. 

Among them, VEGF has also been shown to disrupt integrity of BBB by altering tight junction proteins under ischemic and inflammatory conditions Our hypothesis is that hypoxia induced upregulation of HIF-1α and HIF-1α plays a critical role in BBB damage, we could not exclude that VEGF also contribute to the opening of BBB, but it is the consequence of upregulation of HIF-1α. Reversal effects on HIF1 also have reversal effects on VEGF directly mediated HIF1 expression [20,37,52]. 

Nevertheless, one interesting perspective for our work would be to study the effect of IH on astrocytes. Others mention the direct influence of HIF-1α on astrocytes which allows the formation of “reactive astrocytes” leading to gliosis and are known to have neuroprotective and also neurotoxic effects [53]. All these questions concerning astrocyte behavior are obviously perspectives of this work and will have to be investigated to better understand the consequences of our two induction methods.

The methodological limitations of our study are essentially linked to the adaptation of the system regulating hypoxia for the conditions of IH. The exposure times were intended to mimic IH as encountered in OSA, with brief cycles of hypoxia-reoxygenation. Due to technical constraints, we performed 10 min cycles (5 min normoxia and 5 min hypoxia). In our model, the minimal PO_2_ reached is around 75 mmHg. This is relatively moderated compared to other in vitro models, where PO_2_ < 50 mmHg are applied to cells. It is difficult to compare these values obtained in an in vitro model with arterial PO_2_ values in patients. In humans as in animal models it is very difficult to know precisely the partial pressure of oxygen present during apneic phenomena in the cerebral microvascular blood compartment. Estimates are made from brain tissue PO_2_ measurements which are not very precise. However, we estimate that our model corresponds rather to a moderate SAS. 

Moreover, our results on BBB cellular models are consistent with brain imaging data obtained in apneic patients. These very reproducible results confirm in our opinion in IH an opening of the BBB (at least one functional alteration) involving the protein tight junctions as well as the ABC transporters.

In conclusion, our study aimed to demonstrate the progressive decrease in permeability associated with the activation of HIF-1α in repeated hypoxia stress with a chemical agent. However, we observed an adaptative phenomenon after the third cycle. P-gp and MRP-1 proteins appeared to be key elements in the regulation of BBB dysfunction during IH conditions or upon activation of the HIF-1α pathway. Furthermore, the use of HIF-1α inhibitors improved the BBB disruption and can be considered to be potential therapeutic strategies for hypoxia-induced cognitive disorders.

## 4. Materials and Methods

### 4.1. Chemicals and Reagents

Astrocyte cells (C6, rat astrocyte cell line) and endothelial cells (bEnd.3, mouse endothelial cell line) were from ATCC (Manassas, VA, USA). Cell culture inserts for 24-well plates (transparent PET membrane transwells, 0.4 µm pore size) were obtained from Dominique Dutscher (Alsace, France). Imaging Plate FC, 24-well plates, TC-surface was purchased from Ozyme (Saint Quentin Yvelines, France). The EVOM voltohmmeter system was from World Precision Instruments (Hertfordshire, UK). ZO-1 antibodies were from Life Technologies (cat 40-2200; Saint Aubin, France) and claudin-5 antibodies were from ABCAM (ab15106; Paris, France). P-gp and MRP-1 antibodies were from GeneTex (GTX23364; San Antonio, TX, USA) and Santa Cruz Biotechnology (sc-13960; Dallas, TX, USA), respectively. Standard protein and all compounds for Ringer HEPES buffer, BCECF-AM, probenicid, verapamil, sodium fluorescein (Na-Fl), were from Sigma-Aldrich (St. Quentin Fallavier, France). Rhodamine-123 was from Thermo Fisher Scientific (Eugene, OR, USA). Hydralazine, MTT kit and Dulbecco’s Modified Eagle’s Medium (DMEM) were also from Sigma-Aldrich. Antibodies and reagents for the detection of HIF-1α were products of R&D systems (Lille, France). YC-1 was purchased from VWR International (Fontenay-sous-Bois, France). 

### 4.2. In Vitro BBB Model

The model was composed of two types of cells: endothelial cells (bEnd.3: immortalized mouse brain endothelial cells) and astrocyte cells (C6: rat brain glioma). All cells were used before passage number 30, which corresponds to the time when these cells may begin to lose their properties. Endothelial cells and astrocytes were cultured with DMEM and fetal bovine serum. When the cells reached confluence, they were seeded onto cell culture inserts for 24-well plates with pores of 0.4 µm in diameter, as described previously [27,28]. The bEnd.3 cells were seeded onto the luminal side of the Transwell filter and the C6 cells onto the abluminal side, and these were cocultured for 8 days to obtain the in vitro BBB model. The model was stable for 2 days.

### 4.3. Exposure to IH

To induce IH we used two methods:

#### 4.3.1. Chemical Induction with Hydralazine

We know that hydralazine induces HIF-1α after 2 h of incubation [28] and that HIF-1α returns to basal level after 6 h. Thus, in the present study, when a coculture model was ready, we incubated the luminal side with 100 µM of hydralazine for 2 h, followed by incubation with DMEM for 6 h. This sequence was repeated three times for a total duration of 24 h (Figure 8).

#### 4.3.2. Physical Induction with a Hypoxia Chamber

To induce physical hypoxia, our in vitro BBB model was placed into a system to perform IH as described before [38]. The protocol alternated 5 min at 16% oxygen (i.e., normoxia phase) and 5 min at 2% oxygen (i.e., hypoxia phase) for 2 h. After cells were exposed to 2 h of IH, they were then exposed for 6 h to normoxic conditions (i.e., 16% oxygen). All of this was repeated three times as shown in Figure 8, for a total duration of 24 h. Oxygen concentration in the culture medium was monitored using oxygen sensors (Oxford Optronics), mean PO_2_ cycles in the Transwell inserts were between 78 and 127 mmHg [38].

### 4.4. Transendothelial Electrical Resistance (TEER) Measurements

Evaluation of the tightness of the BBB was performed by TEER measurement. TEER electrodes were recorded with an EVOM resistant meter, with one electrode placed on the luminal side and the other electrode placed on the abluminal side; the endothelial and astrocyte monolayers separated these two electrodes. TEER measurements of Transwells with cells were subtracted by TEER of blank filters, and the result was multiplied by the membrane area to obtain the TEER measurement in Ω·m^2^.

### 4.5. Na-Fl Permeability Measurements

The evaluation of paracellular barrier function of endothelial cells was obtained by measuring the permeability of cells to Na-Fl, a hydrophilic small fluorescent molecule (MW: 376 Da). The medium was removed and cells were washed with Ringer HEPES buffer. Then Na-Fl was diluted to 10 µg/mL with Ringer HEPES buffer, placed onto the luminal side of the insert, and then incubated at 37 °C for 1 h. Concentration of Na-Fl in the abluminal chamber was evaluated using a fluorescence multiwell plate reader at an excitation of 485 nm and an emission of 530 nm wavelengths (FluoroskanAscent^TM^, Thermo Fisher Scientific, Paris, France). The apparent permeability (Papp) is expressed in 10^−6^ cm·s^−1^ and is calculated using the following formula [54]:(1)$$Papp=\frac{Vr}{C0}\times \frac{1}{S}\times \frac{C1}{t}$$

Vr is the volume of medium in the receiver chamber; *C*0 and *C*1 are the concentration of fluorescent compound in the luminal chamber at t_0_ and in the abluminal chamber after t (time), i.e., 3600 s of incubation, respectively, and S is the area of the monolayer.

### 4.6. Whole Cell ELISA Assay

Inserts were washed with 1% BSA in PBS at pH 7.4 and fixed for 20 min at room temperature with 4% paraformaldehyde diluted in 1X PBS at pH 7.4. After fixation, the insert was washed again. To block endogenous peroxidase sites, the insert was treated with 3% H_2_O_2_ in methanol for 30 min, followed by 20% normal goat serum to block unspecific staining. Inserts were incubated with either monoclonal mouse anti-P-gp (2 µg/mL), monoclonal mouse anti-MRP-1 (10 µg/mL), rabbit anti-ZO-1 (4 µg/mL), or rabbit anti-claudin-5 (2 µg/mL) antibodies. Then cells were washed, and peroxidase conjugated anti-mouse or rabbit IgG was added as a secondary antibody for 2 h at room temperature. After washing, tetramethylbenzidine was added for 10 min in the dark at room temperature. The color reaction was measured at 490 nm with a spectrophotometer.

A standard curve based on a series of dilutions of a standard protein was realized to interpret results. Then the antigen concentration in the sample was extrapolated from the linear equation resulting from this standard curve.

#### 4.6.1. Determination of HIF-1α Content

bEnd3 cells were seeded into permeable 96-well plates at a density of 10,000 cells/well and incubated with or without 100 µM hydralazine for 2 h. Cells were fixed with 4% formaldehyde for 10 min immediately or after 6 h of recovery in normoxia without hydralazine. Some wells were pretreated with 6 µM of YC-1 for 30 min before 2 h of hydralazine and fixation. Alternatively, cells were seeded into permeable 96-well plates and exposed to 2 h of IH. Then HIF-1α measurement was studied using a cell-based ELISA assay according to the manufacturer’s instructions. 

#### 4.6.2. Inhibition of HIF-1α Pathway

To study the prevention of BBB impairment by HIF-1 inhibition, we pretreated cells with 6 µM of YC-1 for 30 min before starting hydralazine or IH exposure with or without YC-1. We verified that a concentration of 6 µM inhibited HIF-1 pathway and did not induce cytotoxicity for our different cells.

### 4.7. Statistical Analysis

Statistical analysis was performed using GraphPad software and Kruskal-Wallis test, followed by Dunn’s post hoc applied to multiple comparisons. The differences between means were considered significant when *p*-values were <0.05. The values are expressed as means ± s.e.m.

## Figures and Tables

**Figure 1 ijms-24-05062-f001:**
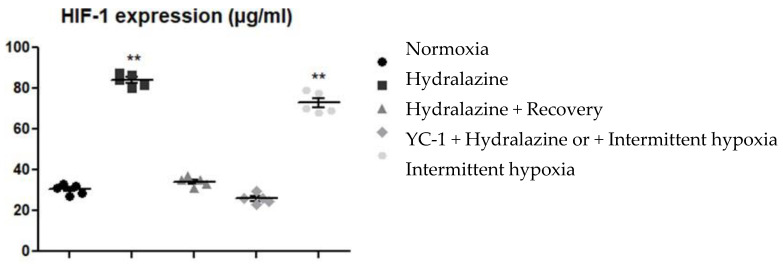
HIF-1α level by b.End3 under different conditions. Treatment with hydralazine for 2 h, hydralazine 2 h + recovery 6 h, pretreatment with YC-1 followed by hydralazine, or physical intermittent hypoxia. Results are presented as mean values ± s.e.m (*n* = 5). ** *p* < 0.01 versus normoxic level.

**Figure 2 ijms-24-05062-f002:**
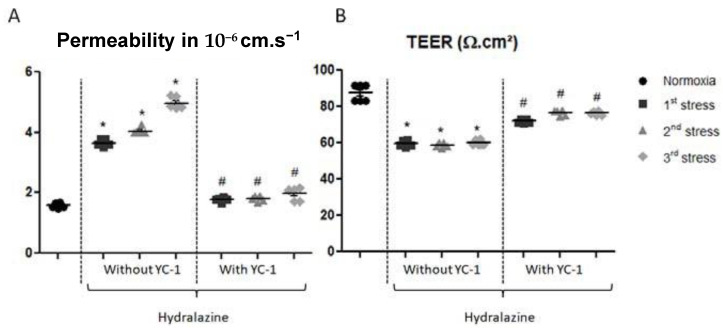
Apparent permeability measurement evaluated by Na-Fl BBB passage (**A**) and transendothelial electrical resistance measurement (TEER) (**B**), after the BBB model was exposed to hydralazine with or without YC-1 treatment, during three cycles of hydralazine/washing. Results are represented as mean value ± s.e.m (*n* = 6). Na-Fl: sodium-fluorescein; BBB: blood-brain barrier. * *p* < 0.05 versus normoxia, # *p* < 0.05 versus without YC-1 treatment.

**Figure 3 ijms-24-05062-f003:**
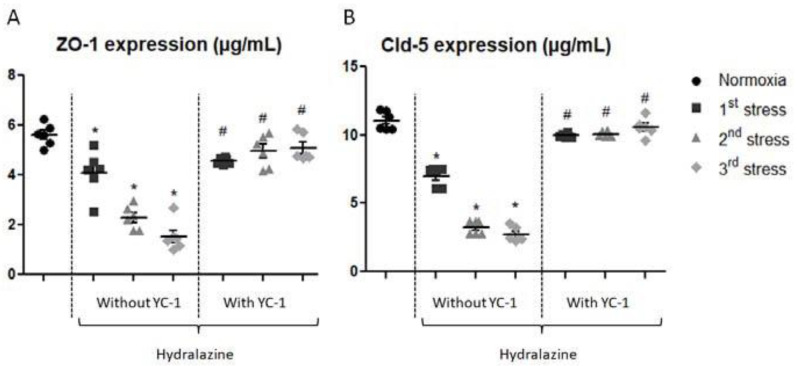
Expressions of ZO-1 (**A**) and claudin-5 (**B**) measured by whole-cell ELISA after exposure of cells to hydralazine, with or without YC-1, during three cycles of hydralazine/washing. Results are represented as mean value ± s.e.m (n = 6). * *p* < 0.05 versus normoxia, # *p* < 0.05 versus without YC-1 treatment.

**Figure 4 ijms-24-05062-f004:**
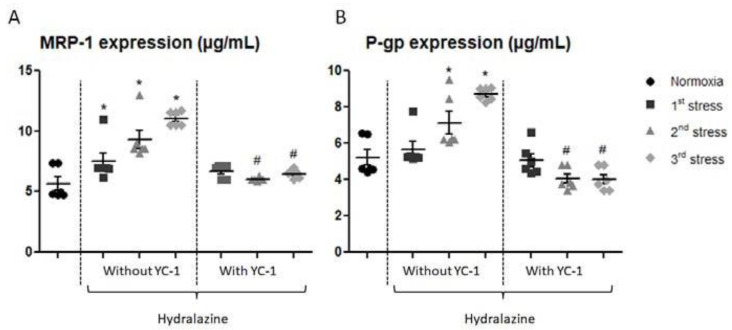
Levels of MRP-1 (**A**) and P-gp (**B**) measured by whole cell ELISA after exposure of cells to hydralazine, with or without YC-1, during three cycles. Results are represented as mean value ± s.e.m (n = 6). * *p* < 0.05 versus normoxia, # *p* < 0.05 versus without YC-1 treatment.

**Figure 5 ijms-24-05062-f005:**
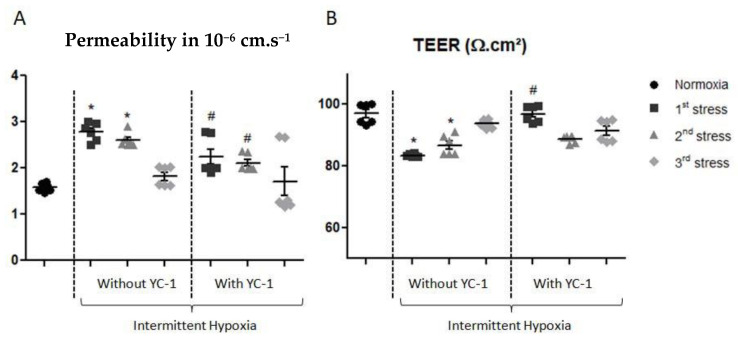
Apparent permeability to Na-Fl (**A**), and transendothelial electrical resistance (TEER) (**B**) after the blood-brain barrier model was exposed to one to three cycles of intermittent hypoxia with or without YC-1 treatment. Results are represented as mean value ± s.e.m (n = 6) Na-Fl: sodium-fluorescein. * *p* < 0.05 versus normoxia, # *p* < 0.05 versus without YC-1 treatment.

**Figure 6 ijms-24-05062-f006:**
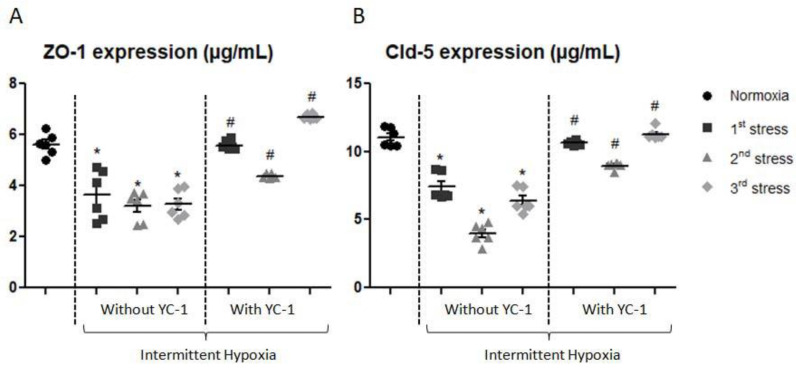
Levels of ZO-1 (**A**) and claudin-5 (**B**) evaluated by whole cell ELISA after exposure of cells to 1 to 3 periods of intermittent hypoxia, with or without YC-1. Results are represented as mean value ± s.e.m (n = 6). * *p* < 0.05 versus normoxia, # *p* < 0.05 versus without YC-1 treatment.

**Figure 7 ijms-24-05062-f007:**
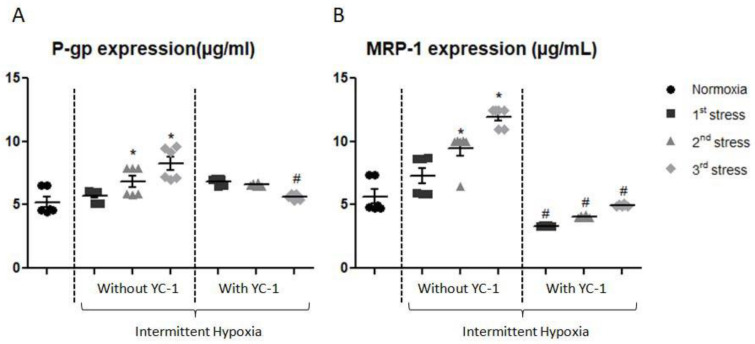
Levels of P-gp (**A**) and MRP-1 (**B**) measured by whole cell ELISA after exposure of cells to one to three periods of intermittent hypoxia, with or without YC-1. Results are represented as mean value ± s.e.m (n = 6). * *p* < 0.05 versus normoxia, # *p* < 0.05 versus without YC-1 treatment.

**Figure 8 ijms-24-05062-f008:**
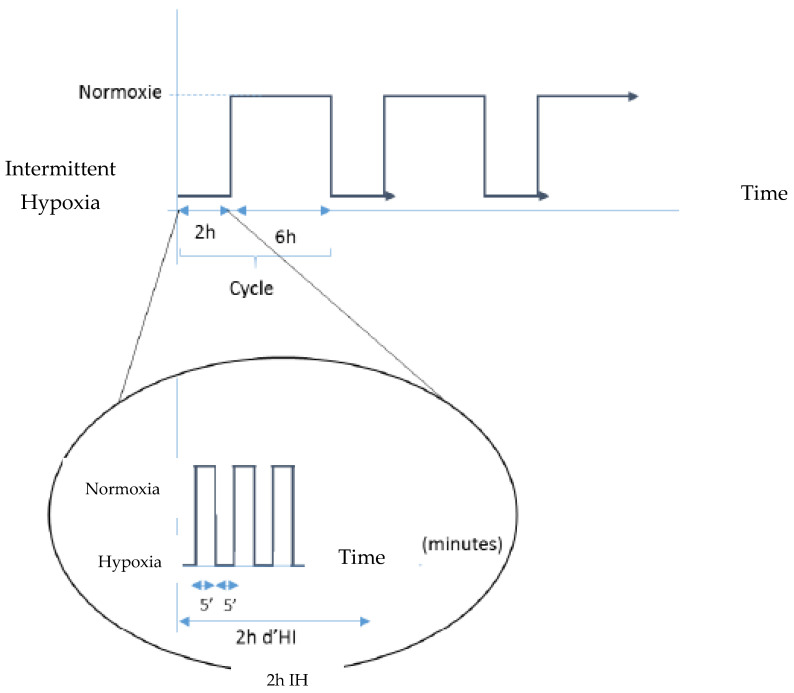
Experimental setups for intermittent hypoxia (IH). Hydralazine cycle corresponded to 2 h with hydralazine and 6 h without hydralazine, repeated 3 times. Physical IH was created by alternating phases of 5 min at 2% oxygen and 5 min at 16% oxygen for 2 h, followed by 6 h of normoxia at 16% oxygen, with the whole process was repeated three times for a total duration of 24 h.

## Data Availability

Not applicable.

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
