# Peer review of "Loss of Blood-Brain Barrier Integrity in an In Vitro Model Subjected to Intermittent Hypoxia: Is Reversion Possible with a HIF-1α Pathway Inhibitor?"

_ijms, 2023, doi:10.3390/ijms24055062_

Round 1
Reviewer 1 Report
Cloé et al. examined the contributions of HIF-1 to intermittent hypoxia-induced blood brain barrier (BBB) disruption in a cerebrovascular endothelium – astrocyte coculture BBB model. Cyclic hypoxia- reoxygenation, or the HIF-1a stabilizer hydralazine, increased BBB permeability and HIF-1 content while lowering contents of the intercellular junction proteins ZO-1 and claudin-5, in a manner suppressed by the HIF-1 inhibitor YC-1. Interestingly, IH and hydralazine increased contents of the membrane transporters multidrug resistance protein-1 (MRP-1) and P-glycoprotein (P-gp); the increased contents of these proteins coincided with attenuation of IH- and hydralazine-induced BBB disruption, and were blunted by YC-1.
This study identifies a potentially adverse effect of HIF-1, namely, the loss of blood brain barrier integrity. Although the present findings are contrary to the well-recognized neuroprotective effects of HIF-1-activated gene expression [Ashok et al., Clin Exp Pharmacol Physiol 2017;44:327 (PMID 28004401); Leu et al., Neurosignals 2019;27:50-61 (PMID 31860206)], they underscore the potential detriments of HIF-1 over-activation which has been implicated in ischemic stroke and neurodegenerative disorders [cf. Chen et al., Curr Neuropharmacol 2022;20:1651 (PMID 34325641); He et al., Front Immunol 2021;12:801985 (PMID 34966392)] and which may limit HIF-1’s neuroprotective capabilities. The present study provides important information regarding the mechanisms of BBB disruption imposed by IH resembling obstructive sleep apnea.
There are a few comments for the authors to consider:
Although VEGF-activated angiogenesis may foster remodeling and recovery of brain infarct [Zhang & Zhao, Am J Chin Med 2014;42:61-77 (PMID 24467535], the authors discuss (lines 444-449) the possibility that excessive VEGF may contribute to the BBB disruption. Thus, the effects of the IH protocol and hydralazine on VEGF content will be of particular interest to readers. The authors are strongly encouraged to report VEGF contents in the different groups, which will provide crucial insights on the mechanism of IH-induced BBB disruption.
The terms “expression” and “overexpression” are used throughout the manuscript (69 times, by my count) to describe the amounts of the analyte proteins. However, “expression” and “overexpression” are undefined, unit-less terms. In the literature, “expression” is used interchangeably to describe the amount (abundance) of messenger RNA, or the activation of a gene, or the content or concentration of a particular protein, or even protein phosphorylation. In this manuscript, “expression” designates the amounts of the proteins of interest. Please use more specific terminology, e.g. “content” or “concentration” of the analyte proteins.
In Figures 2, 4, 5, 7 and 8, “expression” of the analyte proteins is assigned the units of concentration, microgram/ml. But to what does “ml” refer? Is it the volume of incubation medium? The volume of ELISA buffer? Neither of these volumes are meaningful, because the concentration of the analyte protein will depend on the amount of cellular material; thus, differences in cell mass among the wells, e.g. due to cytotoxic effects of IH and/or hydralazine, will alter the amount of analyte protein independent of the targeted effects of the treatments. To account for variations in cell mass, the protein contents must be normalized per microgram (or milligram) of total protein, not per milliliter.
Line 82: Should “advice” be “a device?”
Line 102: Start a new paragraph here.
Line 161: “references cited by Minoves et al.”
Line 166: Were the PO2 values (78-127 mmHg) detected in normoxia, hypoxia or both? In the latter case, it would be surprising that exposure to 2% O2 did not lower the PO2 further. How do these values compare with arterial PO2 values seen in OSA patients?
Line 183: Permeability of cells (implies changes in the plasma membrane)? Or permeability of endothelium (implies changes in the paracellular BBB)?
Lines 215-216: Were proteins extracted from the cells? If so, how?
Lines 245-247 vs. Figure 3A: in the text the units are cm×s-1, while the figure shows 10-6 cm×s-1. Which is correct?
Lines 259-264: Please show only one digit to the right of the decimal.
Lines 347-348 state that HIF-1 expression would contribute to the loss of brain permeability. I think you mean that HIF-1 expression contributes to the increase in brain permeability.
Lines 382-386: Yeh et al. reported ZO-1 disorganization was associated with increased (not decreased) permeability.
Lines 423-424: By “original” do you mean to say that this report is the first evidence of MRP-1 induction by IH?
Please include VEGF on the list of abbreviations.
Author Response
We would like to thank you for reviewing this article and for the very good points that were raised. Please find attached a revised version of our manuscript with changes being highlighted, and a clean revised manuscript.
In accordance with the Reviewer’s comments, the manuscript has been revised with some complementary information added to be in line with the relevant remarks raised by the Reviewer. The other comments raised by the Reviewer have also been answered. We hope that our responses to the comments as well as the modifications brought to the manuscript will be satisfactory.
Please see the attachment
Cloé et al. examined the contributions of HIF-1 to intermittent hypoxia-induced blood brain barrier (BBB) disruption in a cerebrovascular endothelium – astrocyte coculture BBB model. Cyclic hypoxia- reoxygenation, or the HIF-1a stabilizer hydralazine, increased BBB permeability and HIF-1 content while lowering contents of the intercellular junction proteins ZO-1 and claudin-5, in a manner suppressed by the HIF-1 inhibitor YC-1. Interestingly, IH and hydralazine increased contents of the membrane transporters multidrug resistance protein-1 (MRP-1) and P-glycoprotein (P-gp); the increased contents of these proteins coincided with attenuation of IH- and hydralazine-induced BBB disruption, and were blunted by YC-1.
This study identifies a potentially adverse effect of HIF-1, namely, the loss of blood brain barrier integrity. Although the present findings are contrary to the well-recognized neuroprotective effects of HIF-1-activated gene expression [Ashok et al., Clin Exp Pharmacol Physiol 2017;44:327 (PMID 28004401); Leu et al., Neurosignals 2019;27:50-61 (PMID 31860206)], they underscore the potential detriments of HIF-1 over-activation which has been implicated in ischemic stroke and neurodegenerative disorders [cf. Chen et al., Curr Neuropharmacol 2022;20:1651 (PMID 34325641); He et al., Front Immunol 2021;12:801985 (PMID 34966392)] and which may limit HIF-1’s neuroprotective capabilities. The present study provides important information regarding the mechanisms of BBB disruption imposed by IH resembling obstructive sleep apnea.
There are a few comments for the authors to consider:
Although VEGF-activated angiogenesis may foster remodeling and recovery of brain infarct [Zhang & Zhao, Am J Chin Med 2014;42:61-77 (PMID 24467535], the authors discuss (lines 444-449) the possibility that excessive VEGF may contribute to the BBB disruption. Thus, the effects of the IH protocol and hydralazine on VEGF content will be of particular interest to readers. The authors are strongly encouraged to report VEGF contents in the different groups, which will provide crucial insights on the mechanism of IH-induced BBB disruption.
RESPONSE : Hypoxia-inducible factor 1 (HIF-1), and more specifically HIF-1α, is stabilized in response to hypoxia stress, and stimulates the transcription of genes involved in various biological processes. There are several pathways and it is difficult to determine which one is more implicated. HIF-1α exerts its effects through proteins coded by its downstream genes for example Protease Activated Receptor (PAR-1), Vascular endothelial growth factor (VEGF), and glucose transporter (GLUT). In our previous studies, we have seen that intermittent hypoxia provokes BBB disruption mediated PAR1 expression and reversal studies with PAR1 antagonist diminish BBB permeability linked to HIFalpha expression (1). We also demonstrated that IH induces ROS. Indeed, the production of ROS is known to alter BBB functioning mediated NRF2 transcription factor ((2) (3)). These downstream genes may express differently and exert different functions in different cell types. VEGF has been reported to have different effects on cell and tissue injuries. On the one hand, it might directly counteract the detrimental neurological effects associated with stroke. VEGF also promotes blood-brain barrier (BBB) permeability by altering tight junctions under ischemic and inflammatory conditions. Suppressing VEGF by HIF-1 inhibitors improves BBB permeability as observed by Yeh et al(4). Understanding cell-type dependent effects of HIF-1 will undoubtedly shed new lights on its role on BBB opening and provide potential approaches to promote its beneficial effect and reduce its detrimental function. The limitation of our study is to elucidate by which mechanism but we could not exclude that many pathways were involved as we have demonstrated in our previous papers.
We propose to discuss this point in our discussion.
The terms “expression” and “overexpression” are used throughout the manuscript (69 times, by my count) to describe the amounts of the analyte proteins. However, “expression” and “overexpression” are undefined, unit-less terms. In the literature, “expression” is used interchangeably to describe the amount (abundance) of messenger RNA, or the activation of a gene, or the content or concentration of a particular protein, or even protein phosphorylation. In this manuscript, “expression” designates the amounts of the proteins of interest. Please use more specific terminology, e.g. “content” or “concentration” of the analyte proteins.
Thank you for this remark. We have revised the manuscript to use more appropriate words and avoid the repetition.
In Figures 2, 4, 5, 7 and 8, “expression” of the analyte proteins is assigned the units of concentration, microgram/ml. But to what does “ml” refer? Is it the volume of incubation medium? The volume of ELISA buffer? Neither of these volumes are meaningful, because the concentration of the analyte protein will depend on the amount of cellular material; thus, differences in cell mass among the wells, e.g. due to cytotoxic effects of IH and/or hydralazine, will alter the amount of analyte protein independent of the targeted effects of the treatments. To account for variations in cell mass, the protein contents must be normalized per microgram (or milligram) of total protein, not per milliliter.
Response : Thank you for your remark. Since the proteins are not extracted from cells, the technique of whole cell ELISA employed does not allow a normalization with the protein content. However our method is possible as we have verified that treatments did not produce any cytotoxicity (5,6). In our Whole Cell ELISA, we directly quantified with a standard curve based on a series of dilutions of a standard protein. Then the concentration in the sample cell was extrapolated from the linear equation resulting from this standard curve. The linear curve is expressed in µg per mL, and the results were given in this unit.
Line 82: Should “advice” be “a device?”
Thank you so much we have changed in the text
Line 102: Start a new paragraph here.
We have taken in account this new paragraph
Line 161: “references cited by Minoves et al.”
We have taken in account this mistake
Line 166: Were the PO2 values (78-127 mmHg) detected in normoxia, hypoxia or both? In the latter case, it would be surprising that exposure to 2% O2 did not lower the PO2 further. How do these values compare with arterial PO2 values seen in OSA patients?
Response 127mmHg is the maximum during the normoxic phase of intermittent hypoxia cycles, and 78mmHg is the minimum that we could obtain in the upper part of the transwells (hence in the endothelial cells) during the hypoxic phase. It corresponds to about 10% O2, so it is rather high compared to the 2% of dioxygen in the gas mixture. The transwell represent a diffusion barrier, and also enforces a distance from the bottom of the plate, leading to higher minimal values in the transwell than at the bottom of the permeable plate (we have an increasing gradient of O2 from bottom to top, as described in (7).
It is difficult to compare these values obtained in an in vitro model with arterial PO2 values in patients. Indeed, PO2 in brain arterioles and capillaries of OSA patients is not known and one can expect that it is lower than in big arteries. Nonetheless, we estimate that the frequency of our IH cycles (10min per cycle, 6 cycles per hour) as well as the PO2 of 78mmHg correspond rather to a moderate SAS. It is also relatively moderate compared to other in vitro model, where pO2 < 50mmHg are applied to cells.
A paragraph was added in the discussion to comment this point.
Line 183: Permeability of cells (implies changes in the plasma membrane)? Or permeability of endothelium (implies changes in the paracellular BBB)?
We have measured the paracellular transport with sodium fluorescein
Lines 215-216: Were proteins extracted from the cells? If so, how?
The whole Cell Elisa, we have used provided from R&D systems. This assay is based directly on the HIF expression normalized with cytochrome C expression on the whole cell (without any total protein extraction). The manufacturer instructions noted that the plate using a fluorescence plate reader with excitation at 540 nm and emission at 600 nm. Then read the plate with excitation at 360 nm and emission at 450 nm. The readings at 600 nm represent the amount of total HIF-1a in the cells, while readings at 450 nm represent the amount of total Cytochrome c in the cells. The normalization is made with cytochrome C expression on the whole cell as indicated by the manufacturer (R&D system).
Lines 245-247 vs. Figure 3A: in the text the units are cm×s-1, while the figure shows 10-6 cm×s-1. Which is correct?
The permeability is expressed in 10-6 cm.s-1, all has been corrected in the article
Lines 259-264: Please show only one digit to the right of the decimal.
The corrections were made
Lines 347-348 state that HIF-1 expression would contribute to the loss of brain permeability. I think you mean that HIF-1 expression contributes to the increase in brain permeability.
We have changed with the right sentence ‘increase in brain permeability’
Lines 382-386: Yeh et al. reported ZO-1 disorganization was associated with increased (not decreased) permeability.
We have changed the sentence
Lines 423-424: By “original” do you mean to say that this report is the first evidence of MRP-1 induction by IH?
Our work is original because few studies were interested in MRP1 ABC transporter content We avoided the term original in the text
Please include VEGF on the list of abbreviations.
We have completed
References
- Zolotoff C, Puech C, Roche F, Perek N. Effects of intermittent hypoxia with thrombin in an in vitro model of human brain endothelial cells and their impact on PAR-1/PAR-3 cleavage. Sci Rep. 19 juill 2022;12(1):12305.
- Zolotoff C, Voirin AC, Puech C, Roche F, Perek N. Intermittent Hypoxia and Its Impact on Nrf2/HIF-1α Expression and ABC Transporters: An in Vitro Human Blood-Brain Barrier Model Study. Cell Physiol Biochem Int J Exp Cell Physiol Biochem Pharmacol. 17 déc 2020;54(6):1231‑48.
- Pun PBL, Lu J, Moochhala S. Involvement of ROS in BBB dysfunction. Free Radic Res. avr 2009;43(4):348‑64.
- Yeh WL, Lu DY, Lin CJ, Liou HC, Fu WM. Inhibition of hypoxia-induced increase of blood-brain barrier permeability by YC-1 through the antagonism of HIF-1alpha accumulation and VEGF expression. Mol Pharmacol. août 2007;72(2):440‑9.
- Chatard M, Puech C, Perek N, Roche F. Hydralazine is a Suitable Mimetic Agent of Hypoxia to Study the Impact of Hypoxic Stress on In Vitro Blood-Brain Barrier Model. Cell Physiol Biochem Int J Exp Cell Physiol Biochem Pharmacol. 2017;42(4):1592‑602.
- Chatard M, Puech C, Roche F, Perek N. Hypoxic Stress Induced by Hydralazine Leads to a Loss of Blood-Brain Barrier Integrity and an Increase in Efflux Transporter Activity. PloS One. 2016;11(6):e0158010.
- Minoves M, Morand J, Perriot F, Chatard M, Gonthier B, Lemarié E, et al. An innovative intermittent hypoxia model for cell cultures allowing fast Po(2) oscillations with minimal gas consumption. Am J Physiol Cell Physiol. 1 oct 2017;313(4):C460‑8.

Reviewer 2 Report
This study set out to compare two methods of “intermittent hypoxia” effects on the blood brain barrier (BBB) integrity (chemical stress induced by hydralazine and hypoxia exposure using a hypoxia chamber), performed on an endothelial cell and astrocyte coculture model. The authors demonstrate that the use of both methods resulted in increased Na-Fl permeability, indicating altered BBB integrity. Some adaptation was seen after the 3rd cycle of intermittent hypoxia. Inhibition of HIF-1α (with YC-1) prevented BBB dysfunction after hydralazine.
This is an interesting and well-presented study. I only have some rather minor comments:
A clear hypothesis is lacking but should be stated based on the literature findings discussed in the intro section.
As far as I can judge, the methods seem sound to me but some justification (+ refs) for the hydralazine and hypoxia concentrations used and the exposure times (cycling) applied should be provided. As you intended “to improve knowledge on the post-hypoxic cellular risks associated with OSA”, you may briefly tell whether your model really mimics OSA (intermittent hypoxia patterns).
Overall, the findings are clearly presented and discussed.
A limitation section is missing but should be included.
You may elaborate a bit whether and how your in-vitro findings can be transferred to real-world conditions in OSA patients or high-altitude climbers.
Author Response
We would like to thank you for reviewing this article and for the very good points that were raised. Please find attached a revised version of our manuscript with changes being highlighted, and a clean revised manuscript.
In accordance with the Reviewer’s comments, the manuscript has been revised with some complementary information added to be in line with the relevant remarks raised by the Reviewer. The other comments raised by the Reviewer have also been answered. We hope that our responses to the comments as well as the modifications brought to the manuscript will be satisfactory.
PLEASE SEE ATTACHMENT
This study set out to compare two methods of “intermittent hypoxia” effects on the blood brain barrier (BBB) integrity (chemical stress induced by hydralazine and hypoxia exposure using a hypoxia chamber), performed on an endothelial cell and astrocyte coculture model. The authors demonstrate that the use of both methods resulted in increased Na-Fl permeability, indicating altered BBB integrity. Some adaptation was seen after the 3rd cycle of intermittent hypoxia. Inhibition of HIF-1α (with YC-1) prevented BBB dysfunction after hydralazine.
This is an interesting and well-presented study. I only have some rather minor comments:
A clear hypothesis is lacking but should be stated based on the literature findings discussed in the intro section.
Response : We add this paragraph in the introduction part. One hypothesis is that hypoxia repeated stress has consequences on BBB permeability mediated HIF1 α. In this work, we evaluated the impact of repeated hypoxic stress on an in vitro BBB model, inducing HIF-1α content by using either chemical hydralazine or an IH chamber that creates a cellular hypoxic environment alternating with normoxic environment. Firstly, we evaluated the effect of these stresses on the paracellular pathway of the barrier model via the amount of tight junctions and permeability measurement, which are essential properties. Moreover, we evaluated the metabolic properties of the BBB by studying the abundance of ABC transporters necessary for the efflux of exogenous substances from the endothelial cells. Secondly, we evaluated whether inhibition of the HIF-1 α may prevent opening of the BBB, by reversal studies using 3-(5'-hydroxyethyl-2'-furyl)-1-benzylindazole (YC-1) a molecule previously shown to inhibit the expression of HIF-1 alpha in several other studies and in the context of acute hypoxia.
The aim of our study was to improve knowledge on the post-hypoxic cellular risks associated with OSA, as well as to better understand the cellular and molecular mechanisms that would be involved during repeated hypoxia, such as the HIF-1 pathwayAs far as I can judge, the methods seem sound to me but some justification (+ refs) for the hydralazine and hypoxia concentrations used and the exposure times (cycling) applied should be provided. As you intended “to improve knowledge on the post-hypoxic cellular risks associated with OSA”, you may briefly tell whether your model really mimics OSA (intermittent hypoxia patterns).
Response : Hydralazine also shows a capacity to induce a transient and physiological HIF-1α overexpression by inhibiting PHD activity. In the literature, hydralazine was only used to mimic a hypoxic state in cancer models and endothelial cells, I added references in the introduction section
The exposure times were intended to mimic IH as encountered in OSA, with brief cycles of hypoxia-reoxygenation. Due to technical constraints, we performed 10min-cycles (5min normoxia-5min hypoxia). In our model, the minimal pO2 reached is around 75mmHg. This is relatively moderated compared to other in vitro models, where pO2 < 50mmHg are applied to cells. It is difficult to compare these values obtained in an in vitro model with arterial PO2 values in patients. Indeed, PO2 in brain arterioles and capillaries of OSA patients is not known and one can expect that it is lower than in big arteries. Nonetheless, we estimate that the frequency of our IH as well as the PO2 of 78mmHg correspond rather to a moderate SAS.
A paragraph was added in the discussion to comment this point.
Overall, the findings are clearly presented and discussed.
A limitation section is missing but should be included. You may elaborate a bit whether and how your in-vitro findings can be transferred to real-world conditions in OSA patients or high-altitude climbers.
Response : The methodological limitations of our study are essentially linked to the adaptation of the system regulating hypoxia for the conditions of IH. As discussed above, a paragraph is added in the discussion section. In humans as in animal models it is very difficult to know precisely the partial pressure of oxygen present during apneic phenomena in the cerebral microvascular blood compartment. Estimates are made from brain tissue PO2 measurements which are not very precise. However, our results on BBB cellular models are consistent with brain imaging data obtained in apneic patients. These very reproducible results confirm in our opinion in IH an opening of the BBB (at least one functional alteration) involving the protein tight junctions as well as the ABC transporters.

Round 2
Reviewer 1 Report
No further comments.